# Accuracy of Conventional and Digital Impressions for Full-Arch Implant-Supported Prostheses: An In Vitro Study

**DOI:** 10.3390/jpm13050832

**Published:** 2023-05-15

**Authors:** Noemie Drancourt, Chantal Auduc, Aymeric Mouget, Jean Mouminoux, Pascal Auroy, Jean-Luc Veyrune, Nada El Osta, Emmanuel Nicolas

**Affiliations:** 1UFR d’Odontologie, Centre de Recherche en Odontologie Clinique (CROC), Université Clermont Auvergne, CROC, F-63000 Clermont-Ferrand, Franceemmanuel.nicolas@uca.fr (E.N.); 2CHU Clermont-Ferrand, Service d’Odontologie, F-63003 Clermont-Ferrand, France

**Keywords:** digital scanning, conventional impression, accuracy, angular deviations, distance deviations, implant-supported full prosthesis

## Abstract

Both conventional and digital impressions aim to record the spatial position of implants in the dental arches. However, there is still a lack of data to justify the use of intraoral scanning over conventional impressions for full-arch implant-supported prostheses. The objective of the in vitro study was to compare the trueness and precision of conventional and digital impressions obtained with four intra-oral scanners: Trios 4 from 3Shape^®^, Primescan from Dentsply Sirona^®^, CS3600 from Carestream^®^ and i500 from Medit^®^. This study focused on the impression of an edentulous maxilla in which five implants were placed for implant-supported complete prosthesis. The digital models were superimposed on a digital reference model using dimensional control and metrology software. Angular and distance deviations from the digital reference model were calculated to assess trueness. Dispersion of the values around their mean for each impression was also calculated for precision. The mean distance deviation in absolute value and the direction of the distance deviation were smaller for conventional impressions (*p*-value < 0.001). The I-500 had the best results regarding angular measurements, followed by Trios 4 and CS3600 (*p* < 0.001). The conventional and I-500 digital impressions showed the lowest dispersion of values around the mean (*p*-value < 0.001). Within the limitations of our study, our results revealed that the conventional impression was more accurate than the digital impression, but further clinical studies are needed to confirm these findings.

## 1. Introduction

In recent years, interest in digital technologies has increased and impacted various industries around the world, including dentistry. In particular, the advent of computer-aided design/computer-aided manufacturing (CAD-CAM) systems has led to the development of digital technologies for intraoral impressions in the field of prosthodontics. Obtaining quality digital impressions for conventional fixed prostheses is no longer a problem. However, the quality of impressions for implant-supported complete prostheses is still a concern.

The process of osseointegration between alveolar bone and implant body leaves very little margin for error in the accuracy of the impression. In complete oral rehabilitation with a multi-supported implant prosthesis, the passivity of the framework is an essential requirement for the long-term survival of the implants and the prosthetic restoration [1,2,3]. Passivity is even more important for a screw-retained prosthesis, where stresses are applied during the screwing process in order to align the surfaces of the prosthesis and the implant [4]. Accurate transfer of the three-dimensional relationship of the intraoral implant to the master cast is an essential step in achieving a passive fit [5,6].

Conventional impressions are considered the gold standard in some clinical situations and the most commonly used technique in dentistry [7]. However, the risk of deformations during conventional impression or casting of impressions increases during rehabilitation with implants-supported complete prosthesis. It contributes to a lack of passive adaptation of the framework to the implant. Intraoral acquisition provides a digital model in which the implant replicas are automatically placed. The dental technician can model the prosthesis using CAD software and then machine or print the prosthetic part. Digital impression systems eliminate the error-prone plaster modeling step of conventional impressions, and the impression can be stored as an STL file for an unlimited time. Digital technology can provide greater reliability by eliminating casting stresses and/or dimensional variations experienced by materials during curing or removal [8].

Different capture techniques are used in intraoral scanners. The triangulation technique (for I-500, Cs3600 cameras) aims to assume the measurement of the volume of the object by calculating the difference between the incident and reflected light in contact with the object. This acquisition process requires software with significant computing power and complex algorithms capable of reconstructing the surface in three dimensions. Parallel confocal imaging (for Trios 4 and Primescan cameras) is a technique based on laser and optical scanning of the oral volume (dental, implant, periodontal) to digitally reproduce it. A series of “sections” at different depths of field are obtained and assembled to obtain a three-dimensional representation of the object by reconstruction [9]. Some studies agree that not all scanners are suitable for taking digital impressions for full-arch implant-supported prostheses [10]. An inaccurate impression does not record the true position of the implants and the spatial relationships with the teeth, alveolar ridges and soft tissues [11]. Inadequate accuracy of the impression technique and/or manual steps in the fabrication of the prosthesis may lead to poor prosthetic fit and subsequent technical, mechanical and biological complications [10]. Therefore, there is still a lack of data to justify the use of digital impressions in implant-supported complete prosthesis [7].

The aim of this in vitro study was to determine and compare the trueness and the precision of four intraoral scanners (IOS): Trios 4 from 3Shape^®^, Primescan from Dentsply Sirona^®^, CS3600 from Carestream^®^ and i500 from Medit^®^, and those obtained with conventional impression in a full-arch implant-supported prosthesis. The null hypothesis was that conventional and digital impressions produce casts of similar accuracy.

## 2. Materials and Methods

### 2.1. Design

This in vitro study focused on the impression of an edentulous maxilla in which five implants were placed in the right central incisor, canine and first molar sectors and in the left canine and first molar sectors.

### 2.2. Working Model

In the first step, the maxillary working model was produced with a 3D printer from the digital file of a fully edentulous maxillary arch. The model was printed with the Formlabs Form 2^®^ 3D printer, which uses resin as the printing material (Figure 1a). Naturactis^®^ implant analogs of the Euroteknica^®^ (ETK) brand with a diameter of 3.5 mm were fixed on the model in the right sector of the first molar (#1), canine (#2), central incisor (#3) and in the left sector of the canine (#4) and first molar (#5). They have an internal hexagonal conical connection (Ref. NLA_H35) (Figure 1b). The working model was scanned with 3shape D2000 scanner to obtain the digital reference model. This scanner allows multiline scanning using four 5.0-megapixel cameras with 27 blue LEDs (Figure 2).

Two types of impressions were subsequently evaluated: (i) conventional impressions with elastomers, (ii) digital impressions with four different scanners (Trios 4 from 3Shape^®^, Primescan from Dentsply Sirona^®^, CS3600 from Carestream^®^ and Medit i500 ^®^) (Figure 2).

### 2.3. Conventional Impression

In a second step, the impression transfers were screwed on to the maxillary working model at 5 N·cm before taking the conventional impression (Figure 1c). This allows the spatial position of the implant to be transferred into the impression material for precise repositioning of the implant analog on the working model. The transfers were ETK Naturactis^®^ short transfers with S-Naturactis screws (Figure 1d).

Each of the three calibrated operators took three impressions using a custom open tray and polyether material, Impregum™ Penta™ Soft from 3M^®^ (Figure 1e). The custom impression trays were created digitally using Dental System’s 3D modeling software (3Shape^®^) and the digital file of the previously scanned maxillary working model and were printed with the Formlabs’ Form 2^®^ 3D printer. The Impregum™ pellets (base and catalyst) were inserted into the metal cartridge of the 3M^®^ ESPE Pentamix™ 2 automatic mixer with a single-use mixing tip and a 3M^®^ elastomer syringe. Once the implant replicas were placed, the impressions were cast and scanned with the D2000 laboratory scanner to obtain a digital model for evaluations. A total of nine conventional impressions were obtained.

### 2.4. Digital Impression

In the third step, scanbodies were screwed onto the maxillary working model analogs at 5 N·cm prior to digital impression (Figure 1f). The scanbodies were attached to the implants so the scanner can establish the spatial position of the implant analog in the working model.

Each of the three calibrated operators took three impressions with each of the OISs according to the manufacturers’ instructions [12,13,14]. A total of thirty-six digital impressions (nine per camera) were created. These impressions provided the digital models needed for the evaluation.

Two different scanning methods were applied as suggested by the manufacturers [12,13,14]. The first method is common to the Trios 4, Primescan and i500 scanners. It includes a first scan of the reference digital model without scanbodies. The scan started with the occlusal surface from the right molar sector to the left molar sector, then proceeded to the buccal surfaces of the edentulous ridge in a reverse path and finally to the palatal surfaces of the edentulous ridge from the right molar sector to the left molar sector (Figure 3). The second scan was performed with the scanbodies in place, after the circular cut and implant areas removal.

The second method was performed with the CS 3600 scanner. The first step of the scan was performed without the scanbodies and was identical to the previous method (Figure 3). Indeed, after the circular cutting of the implant areas, one scanbody was placed and selected for scanning. Then, the scanbody was removed and the next one was placed for scanning until all scanbodies were completely scanned on the 3D model.

### 2.5. Comparison of Digital Models

The evaluation criterion for the impressions was accuracy, a combination of precision and trueness (ISO 5725-1) [15]. Trueness is the difference between the mean value and the true value. Precision is the distribution of values around the mean that provides the reproducibility of a measurement.

All STL files corresponding to the digital reference model, the 9 conventional impression models and the 36 digital impression models were saved in the same folder on the USB drive for comparison. To compare the accuracy between the conventional and digital impressions, several measurements were performed using dimensional control and metrology software Geomagic^®^ Control X™ (3DSYSTEM^®^). This software was used to compare two 3D digital models: a reference digital model and a model to be compared by aligning and superimposing them to calculate measurement differences (distances, angles) along a particular axis. The software allowed the process to be fully automated by keeping the reference digital model with the measurements to be completed and in turn replacing the model to be compared with the other selected digital models. All the digital models corresponding to the conventional and digital impressions were automatically compared one by one to the reference digital model and reports were generated for each comparison.

The angular and distance differences between the implants were calculated to obtain the accuracy for each type of impression. A comparison of distances was performed between each scanbody for each model: between 1 and 2, 1 and 3, 1 and 4, 1 and 5, between 2 and 3, 3 and 4 and finally 4 and 5. An angular comparison of each scanbody of the reference digital model with the scanbody corresponding to the model to be compared was also performed. The distance deviations were evaluated in two ways, firstly as an absolute value, to obtain an average of the deviations for each impression, and secondly as a negative or positive raw value, to assess the direction of the deviation. The angular deviation corresponds to the angle between the reference digital model and the analyzed impression vectors. A method comparing the dispersion of the values around their mean for each impression was also applied to assess the precision.

IBM SPSS version 28.0 was used to analyze the data. The level of significance was set at *p*-value ≤ 0.05. Repeated measures analyses of variance followed by Bonferroni multiple comparisons tests were applied for statistical comparisons.

## 3. Results

### 3.1. Trueness

#### 3.1.1. Distances Deviation in Absolute Value According to the Impression Type

The mean absolute distance deviation was significantly lower for conventional impression and elevated for CS3600 digital impression (*p* < 0.001) (Figure 4).

#### 3.1.2. Direction of the Distance Deviations According to the Impression Type

The direction of the distance deviation from the digital reference model was significantly different between impressions (*p*-value < 0.001). The conventional impressions and the Trios 4 had mostly positive deviations and the distance deviation was significantly smaller with conventional impression (*p*-value < 0.001) (Figure 5). The results between calibrated operators were not significant (*p* > 0.05).

#### 3.1.3. Angular Deviations According to the Type of Impression

The mean angular deviations were significantly different between impressions (*p* < 0.001). They were smaller with I-500 digital impression, followed by Trios 4 and CS3600 (Figure 6). The results between calibrated operators were not significant (*p* > 0.05).

### 3.2. Precision

The average dispersion of distance values around their mean for each impression is displayed in Table 1. The I-500 has the lowest mean dispersion, followed by conventional impression (*p*-value < 0.001). The Trios 4 and CS 3600 have the highest dispersion, indicating the lowest precision (Table 1).

The average dispersion of angular values around their mean for each impression is shown in Table 2. The I-500, CS3600, Trios 4 and conventional impression have the lowest dispersion (highest precision) and the Primescan has the highest dispersion (lowest precision).

## 4. Discussion

Our study aims to compare the accuracy of four intraoral scanners and one conventional impression for full-arch implant-supported prostheses. The first null hypothesis that conventional and digital impressions will produce casts of similar trueness was rejected. As a result, the mean absolute distance deviation and the direction of the distance deviation from the reference digital model were reduced for the conventional impression. Similarly, the I-500 had the lowest angular deviation, followed by the Trios 4 and the CS3600. The second null hypothesis that conventional and digital impressions will produce casts of similar precision, was also rejected. Thus, the conventional and I-500 impressions showed the lowest dispersion of values around their means. The calculated power of the post tests was greater than 80%, indicating that the sample was sufficiently powerful to detect a difference between the groups.

In terms of trueness, the conventional impression provided the best accuracy for distances, with a mean distance deviation and standard error from the reference model of 132.3 ± 21 μm. The new IOS Primescan and Trios 4 performed better than the older CS3600 and i500 in terms of distance deviation. For an edentulous model, the cameras can easily confuse scanbodies of the same shape, forcing the area to be removed and the camera passageway to be repeated at that level. The possible confusion between the scanbodies in the case of edentulous cast is due to the fact that there are few anatomical shapes and features that allow the virtual model building software to find its position.

A study conducted in 2017 showed that an average distance deviation of 50 to 100 μm and a maximum angular deviation of 1° were required to ensure framework passivity in implant-supported complete prostheses [16]. Our results showed that the distance deviation of the four digital impressions was between 170 and 270 μm. Thus, digital impressions seem to have an accuracy that is not compatible with the passivity of the framework in an implant-supported complete prosthesis. Although the conventional impression had the smallest distance deviation, it was greater than the accepted values (132 μm) and thus possessed a trueness that could lead to framework non-passivity.

For angular deviations, our results revealed that three of the scans (I-500, Trios 4 and CS3600) performed better, but both conventional and digital impressions were below the 1° value.

In terms of precision, the distances between the scanbodies affect the dispersion of values from the mean. The greater the distance from the starting point (scanbody 1), the greater the average dispersion. The average dispersion increases from distance 1–2 to distance 1–5, and from distance 2–3 to distance 4–5. Since the distance between scanbody 2 and scanbody 3 was the shortest on the baseline model, the average dispersion was the smallest. For the angular differences between each scanbody and the reference, the results showed smaller average deviations, ranging from 0.212° for the i500 scanner, 0.317° for conventional impression to an increase in average deviation of 0.966° for the Primescan.

Combining the results of deviations from the average for distances and angles, the i500 scanner achieved the best precision, followed by conventional impression. However, the dispersion of values from the angular mean was high for Trios 4 (233.6 μm) and the dispersion of values around the distance mean was high for Primescan 0.966°.

Implant ankylosis imposes several constraints on the implant-supported prosthesis [16,17,18,19]. Therefore, precision and trueness are crucial outcomes [6,20]. In this context, the digital impression has many advantages [17]: (i) the suppression of impression trays and materials reduces the risks related to incomplete curing and deformations; (ii) the reduction in errors related to laboratory processing (casting, demolding and transfer placement) ensures the stability of the impression; (iii) real-time processing of information and ease of reinterventions (reuse of the computer file or possibility of partial modification); (iv) improved patient comfort (less nausea) and the possibility to interrupt and continue the impression process at any time without losing the information already acquired, and patients prefer digital impression [21]; (v) the preservation of the virtual model, which, under appropriate clinical conditions, provides the possibility to remake the prosthetic element without patient intervention; (vi) the availability of libraries of theoretical scanbody morphologies suitable for different implant types; (vii) the use of the IOS saves time and (viii) improved communication with the laboratory: fast delivery without a carrier, exchange with the prosthetist before machining, help in choosing the shade and saving time [22].

Digital impression also has its limitations. Many factors can interfere with the scanning of a dental arch. These may include operator or equipment errors (lack of calibration), the nature of the object to be scanned [23] or external disturbances, such as the lighting in the clinic [24,25]. The optical properties of the scanned elements (reconstruction materials and prosthesis) contribute to alter the acquisitions [26].

Conventional impressions are considered the gold standard in some clinical situations and the most commonly used technique in dentistry [7]. However, it represents one of the weakest links in the prosthetic design. Conventional impressions are associated with limitations, such as ongoing costs, patient discomfort, the need for well-fitted impression trays and the need to cast with dental stone. In addition, their quality depends on material handling, deformation of the impression and stone material and capture of all intraoral tissues [27].

Our results showed that conventional impression was the most accurate. The four scanners provided very different results. This study confirmed the results of a previous study showing that digital impression is accurate but not yet faithful enough to be used routinely in implant-supported complete prostheses. Indeed, the four intraoral scans showed great variability [28].

It is worth noting that the results of this in vitro study do not presume the clinical validity of the impressions. Clinical studies are needed to evaluate the accuracy of these impressions in an implant-supported complete prosthesis as many important parameters (mouth opening, presence of blood or saliva, anatomical obstacles, etc.) may affect their accuracy. However, in vitro experiments have the advantage of limiting confounding parameters and evaluating all IOS under the same conditions. In this context, it is important to follow the manufacturer’s instructions, calibrate accurately, and change the tip regularly to take full advantage of the IOS’s performance. In addition, the use of the mesh/mesh (virtual model) method to evaluate all the impressions may also be a limitation of our study. Meshes are surface reconstructions, and thus geometric approximations of the scanned model, which may induce errors in the calculation of distances between scanbodies. However, in the case of implant prostheses, the first step in CAD is to replace the scanbody meshes with the corresponding scanbody library file. This is a geometrically perfect file (NURBS file, Non-uniform Rational B-Splines). The use of these NURBS files allows to obtain more reliable linear distances.

## 5. Conclusions

Within the limits of the in vitro study, our results revealed that the conventional impression was more accurate than the digital impression, but further clinical studies are needed to confirm these results. Nevertheless, the continuous progress of intraoral scanning technologies and the development of new acquisition processes might allow optical impressions to extend its indications in implantology and match or even surpass conventional impressions for implant-supported complete prosthesis.

## Figures and Tables

**Figure 1 jpm-13-00832-f001:**
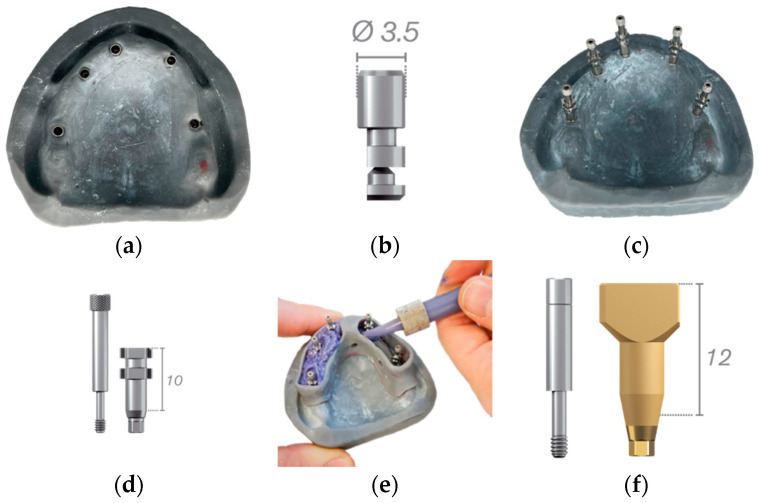
(**a**): Maxillary working model. (**b**): Implant analog *ETK Naturactis (Lyrashop).* (**c**): Impression transfers screwed on maxillary working model. (**d**): Short ETK Naturactis^®^ transfers with S-Naturactis screws. (**e**): Custom open tray with polyether material. (**f**): Scanbody and ETK screw (Lyrashop).

**Figure 2 jpm-13-00832-f002:**
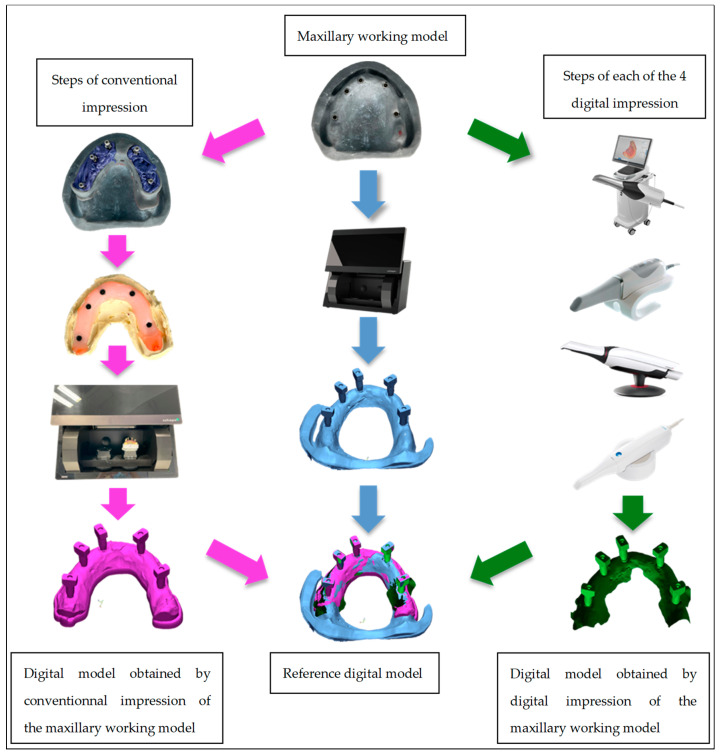
Diagram illustration of the in vitro study.

**Figure 3 jpm-13-00832-f003:**
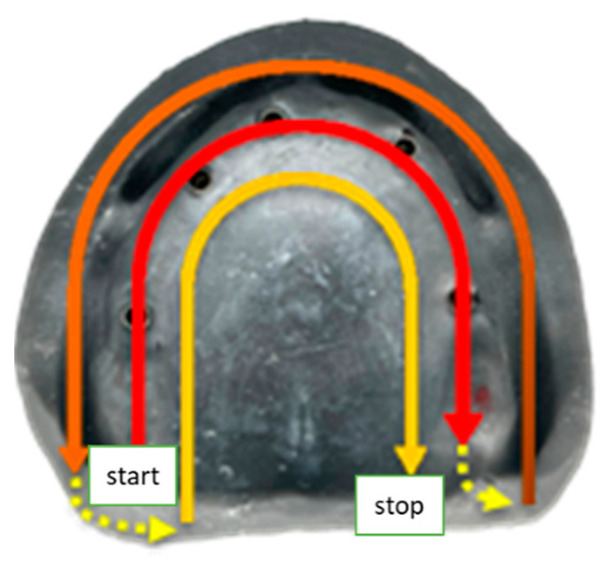
Directions of the scanning sequence.

**Figure 4 jpm-13-00832-f004:**
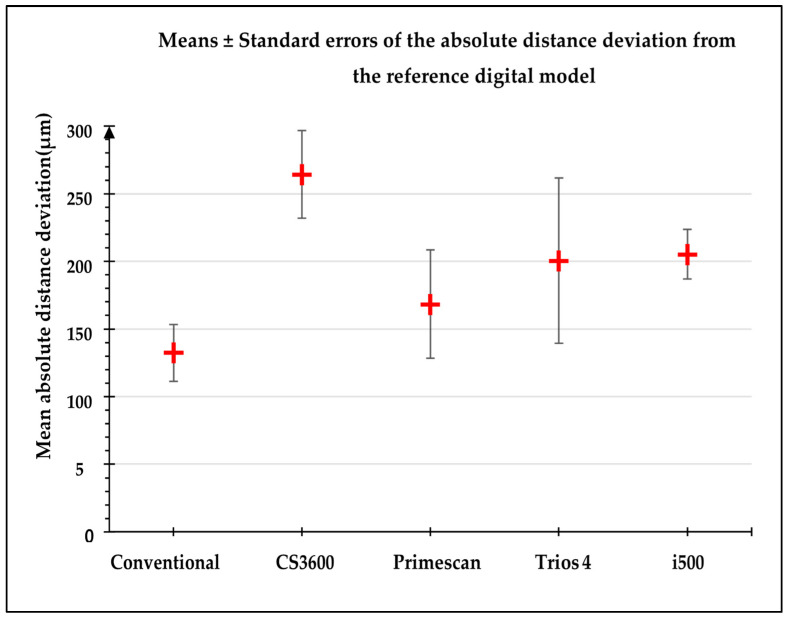
Mean absolute distances deviation (μm) for the different types of impressions.

**Figure 5 jpm-13-00832-f005:**
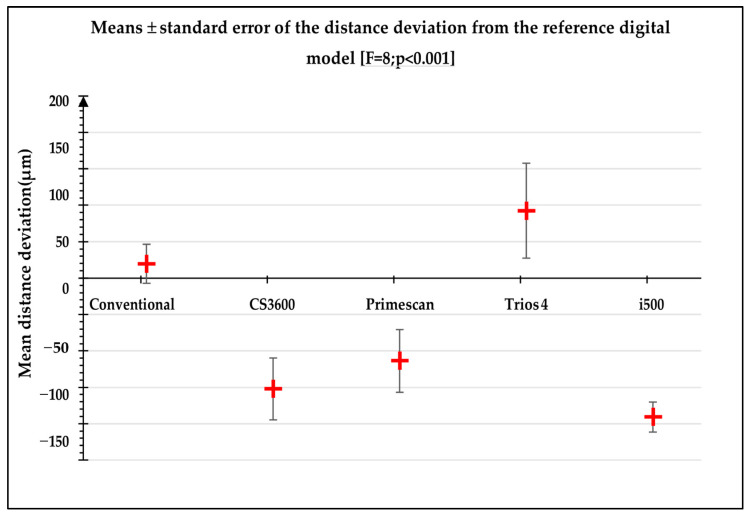
Direction of the distance deviation (μm) for the different types of impressions.

**Figure 6 jpm-13-00832-f006:**
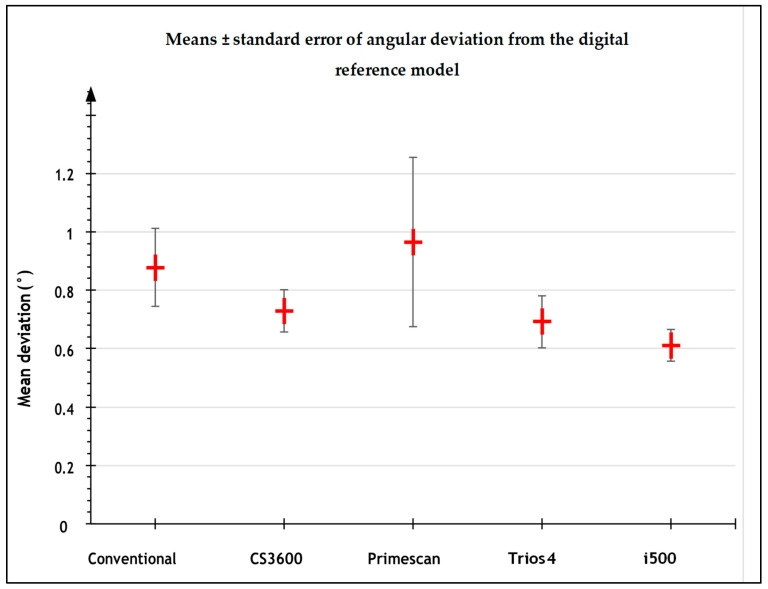
Mean angular deviation (^o^) from the reference digital model for the different types of impressions.

**Table 1 jpm-13-00832-t001:** Average dispersion of values around the mean (distances in μm).

Distances between Scanbodies	Conventional	CS3600^®^	Primescan^®^	i500^®^	Trios 4^®^
**1–2**	66.6	179.6	25.7	109.8	167.8
**1–3**	62.6	168.7	35.7	118.7	149.8
**1–4**	88.0	165.7	204.2	116.5	229.1
**1–5**	173.7	260.8	82.7	114.6	142.5
**2–3**	47.2	85.1	35.8	35.2	18.4
**3–4**	48.7	234.2	381.9	59.3	710.4
**4–5**	399.0	335.9	282.0	111.3	217.4
**Mean ± SD**	126.5 ± 128	204.3 ± 81	149.7 ± 142	95.1 ± 34	233.6 ± 221

**Table 2 jpm-13-00832-t002:** Average dispersion of values from the mean (angles in degrees).

Angles	Conventional	CS3600^®^	Primescan^®^	i500^®^	Trios 4^®^
**Scanbody 1**	0.190	0.269	0.266	0.310	0.283
**Scanbody 2**	0.077	0.266	0.304	0.138	0.210
**Scanbody 3**	0.161	0.236	0.327	0.115	0.231
**Scanbody 4**	0.338	0.235	2.413	0.140	0.230
**Scanbody 5**	0.818	0.571	1.517	0.355	0.661
**Mean ± SD**	0.317 ± 0.296	0.315 ± 0.144	0.965 ± 0.966	0.212 ± 0.112	0.323 ± 0.191

## Data Availability

Not applicable.

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
