# Peer review of "Accuracy of Conventional and Digital Impressions for Full-Arch Implant-Supported Prostheses: An In Vitro Study"

_jpm, 2023, doi:10.3390/jpm13050832_

Round 1
Reviewer 1 Report
The topic is current and of high interest for the user of these processes.
We suggest to add the following points.
a) for method, better description of the scan path for each different scanner. Were the respective recommendations of the manufacturers followed here.
Was the jaw scanned completely with the entire palate?
In the discussion and summary, I would still like to see a clinical evaluation. Is the conclusion that conventional impression taking must still be recommended at the moment?
Author Response
2.1- for method, better description of the scan path for each different scanner. Were the respective recommendations of the manufacturers followed here.
The exact steps of the procedure were rewritten in the Material and Methods section as requested by the reviewer. The scan path for each scanner were described. Each calibrated operator took impressions with each of the OISs in accordance with the manufacturers' instructions as described on the website of each scanner. These references were added in the manuscript.
- A Quick Start Guide for the First Scan Using the I500 Available online: https://support.medit.com/hc/en-us/articles/360042009112-A-quick-start-guide-for-the-first-scan-using-the-i500
- Carestream DENTAL CS 3600 User And Installation Manual (Page 8 of 22) | ManualsLib Available online: https://www.manualslib.com/manual/2313009/Carestream-Dental-Cs-3600.html?page=8
- Caméra Primescan | Dentsply Sirona France Available online: https://www.dentsplysirona.com/content/dentsply-sirona/fr-fr/decouvrez-nos-produits/impression-numerique/primescan.html
Each of the three calibrated operators took three impressions with each of the OISs according to the manufacturers' instructions [12–14]. A total of thirty-six digital impressions (nine per camera) were made. These impressions provided the digital models needed for the evaluation.
Two different scanning methods were applied as suggested by the manufacturers [12–14]. The first method is common to the Trios 4, Primescan and i500 scanners. It includes a first scan of the reference digital model without scan bodies. The scan started with the occlusal surface from the right molar sector to the left molar sector, then proceeded to the buccal surfaces of the edentulous ridge in a reverse path and finally to the palatal surfaces of the edentulous ridge from the right molar sector to the left molar sector (Figure 8). The second scan was performed with the scan bodies in place, after the circular cut and implant areas removal.
The second method was performed with the CS 3600 scanner. The first step of the scan was performed without the scan bodies and was identical to the previous method (Figure 8). Indeed, after the circular cutting of the implant areas, one scan body was placed and selected for scanning. Then, the scan body was removed and the next one was placed for scanning until all scan bodies were completely scanned on the 3D model.
Figure 8: Directions of the scanning sequence
2.2- Was the jaw scanned completely with the entire palate?
In this study, the palate was not scanned as this would have added image files and thus potential additional errors.
2.3- In the discussion and summary, I would still like to see a clinical evaluation.
For the abstract, we write:
Within the limitations of our study, our results revealed that the conventional impression was more accurate than the digital impression, but further clinical studies are needed to confirm these findings.
For the discussion, we write:
It is worth noting that the results of this in vitro study do not presume the clinical validity of the impressions. Clinical studies are needed to evaluate the accuracy of these impressions in implant-supported complete prosthesis, as many important parameters (mouth opening, presence of blood or saliva, anatomical obstacles, ...) may affect their accuracy. However, in vitro experiments have the advantage of limiting confounding parameters and evaluating all IOS under the same conditions. In this context, it is important to follow the manufacturer's instructions, calibrate accurately, and change the tip regularly to take full advantage of the IOS's performance. In addition, the use of the mesh/mesh (virtual model) method to evaluate all the impressions may also be a limitation of our study. Meshes are surface reconstructions, and thus geometric approximations of the scanned model, which may induce errors in the calculation of distances between scan bodies. Though, in the case of implant prostheses, the first step in CAD is to replace the scan body meshes with the corresponding scan body library file. This is a geometrically perfect file (NURBS file, Non-uniform Rational B-Splines). The use of these NURBS files allows to obtain more reliable linear distances.
Is the conclusion that conventional impression taking must still be recommended at the moment?
In the conclusion section, we concluded that our in vitro results revealed that the conventional impression was more accurate than the digital impression, but further clinical studies are needed to confirm these results. Nevertheless, the continuous progress of intraoral scanning technologies and the development of new acquisition processes may allow optical impressions to extend its indications in implantology and match or even surpass conventional impressions for implant-supported complete prosthesis.
Within the limits of the in vitro study, our results revealed that the conventional impression was more accurate than the digital impression, but further clinical studies are needed to confirm these results. Nevertheless, the continuous progress of intraoral scanning technologies and the development of new acquisition processes may allow optical impressions to extend its indications in implantology and match or even surpass conventional impressions for implant-supported complete prosthesis.

Reviewer 2 Report
Manuscript on a current, important topic that requires frequent evaluations due to the progress observed in the development of new versions of intraoral scanners.
Some questions and suggestions:
There is a need for English correction, very carefully!
Abstract:
I understand that the conclusion presented in the Abstract is not adequate! I suggest reviewing.
Introduction
Could you say something about the different capture methods used, even specifying what each tested scanner does. What are the biggest difficulties?
And I suggest establishing research hypotheses.
Methods
Wouldn't it be better to call this section Materials and Methods?
Your Figure 1 is bad, it needs to be more explanatory and have better resolution!
The description in the text should also be revised.
I was in doubt about the exact steps in the three illustrated sequences.
Subheading 2.3: Is the "5 N.m" correct? That would be the equivalent of 500 N.cm of torque!!!
For the elastomer impression, were the transfers joined together? If so, with what material? If not, why not?
Did you do any sample calculations? Please, explain.
You said that "...Each calibrated operator made three impressions with each of the OISs according to the manufacturers' recommendations." What exactly are these recommendations? Are they the same for each scanner?
I am referring here more to the scanning sequence, in terms of directions and not the steps described below.
What are implants 1, 2, 3, 4 and 5? Not identified!
Results
According to what is illustrated in Figure 1, you superimposed 3 digital models, right?
So, it is necessary to explain better what they obtained as a result.
What, exactly, do you call a "reference"?
Discussion
The Discussion should be rewritten contemplating some of the suggestions made above and making considerations about important points that generate doubts.
For example: what is the impact of model scans for the analyzes being done? Can't benchtop scanning also have its own problems?
In addition, it deepens the considerations about extraoral scanning with intraoral scanners, which are quite different things.
If research hypotheses are formulated, each one should be accepted or rejected and discussed.
Regarding the limitations, I also understand that they can be better evaluated.
Conclusion
Regarding the conclusion, as presented it looks more like a discussion paragraph, besides not corresponding to what appears in the Abstract.
References
Reference #15 is too old (1991)!
Author Response
- ANSWERS TO THE COMMENTS AND SUGGESTIONS OF THE REVIEWER 1
- There is a need for English correction, very carefully!
Thank you for the comment. The manuscript was reviewed for English correction.
Abstract
- I understand that the conclusion presented in the Abstract is not adequate! I suggest reviewing.
We have adjusted the conclusion presented in the Abstract as requested by the reviewer:
Within the limitations of our study, our results revealed that the conventional impression was more accurate than the digital impression, but further clinical studies are needed to confirm these findings.
Introduction
- Could you say something about the different capture methods used, even specifying what each tested scanner does. What are the biggest difficulties?
We added in the Introduction section a paragraph about the different capture methods and difficulty.
Different capture techniques are used in intraoral scanners. The triangulation technique (for I-500, Cs3600 cameras) aims to assume the measurement of the volume of the object by calculating the difference between the incident and reflected light in contact with the object. This acquisition process requires software with significant computing power and complex algorithms capable of reconstructing the surface in three dimensions. Parallel confocal imaging (for Trios 4 and Primescan cameras) is a technique based on laser and optical scanning of the oral volume (dental, implant, periodontal) to digitally reproduce it. A series of “sections” at different depths of field are obtained and assembled to obtain a three-dimensional representation of the object by reconstruction [9]. Some studies agree that not all scanners are suitable for taking digital impressions for full-arch implant-supported prostheses [10]. An inaccurate impression does not record the true position of the implants and the spatial relationships with the teeth, alveolar ridges and soft tissues [11].
- I suggest establishing research hypotheses.
We added the research hypotheses as requested by reviewers in the Introduction section.
Our research hypothesis was that the impression obtained with intraoral scans is more accurate and precise than that obtained with the conventional impression for full-arch implant prostheses.
Methods
- Wouldn't it be better to call this section Materials and Methods?
We called this section Material and Methods.
- Your Figure 1 is bad, it needs to be more explanatory and have better resolution!
Thank you for the comment. We have modified Figure 1 with a better resolution. This Figure was better described and illustrated in the material and methods section.
- The description in the text should also be revised. I was in doubt about the exact steps in the three illustrated sequences.
The exact steps of the procedure were rewritten in the Material and Methods section as requested by the reviewer.
- Subheading 2.3: Is the "5 N.m" correct? That would be the equivalent of 500 N.cm of torque!!!
Thank you for your attention. The error concerning the torque has been corrected as 5 N.cm.
- For the elastomer impression, were the transfers joined together? If so, with what material? If not, why not?
The reviewer makes a valid point.
For impression, we used the “open tray” technique, in which the transfers remain enclosed in the impression material after the impression has been removed from the patient’s mouth.
There is no evidence to date that impressions made using the unsplinted technique for open tray impression, lead to a compromised prosthesis and poorer long-term clinical outcomes.
The decision to splint or not splint impression copings is ultimately one of personal preference of the clinician since additional materials and extra chair time are required when the splinted technique is utilized. Also, some authors have emphasized the risk of rotation of the impression coping in the bulk of impression material when attaching the analog.
References :
- Del’Acqua MA, Arioli-Filho JN, Compagnoni MA, Mollo F de A Jr. Accuracy of impression and pouring techniques for an implant-supported prosthesis. Int J Oral Maxillofac Implants 2008;23:226-36.
- Carr AB. Comparison of impression techniques for a five-implant mandibular model. Int J Oral Maxillofac Implants 1991;6:448-55
The use of an adhesive on impression copings leads to irreversible deformation of the interface at torque stresses well below the adhesive bond threshold of the same materials used without an adhesive.
it was shown that irreversible rotations between the impression coping and the bulk of impression material occur for lower torque values when using an adhesive than when no adhesive is used.
Reference:
- Auroy P, Nicolas E, Bedouin Y. Torque resistance of impression copings after direct implant impression: An in vitro evaluation of impression materials with and without adhesive. J Prosthet Dent. 2017 Jan;117(1):73-80. doi: 10.1016/j.prosdent.2016.05.002. Epub 2016 Jul 28. PMID: 27475917.
- Did you do any sample calculations? Please, explain.
Thank you for the comment.
For this study, the power post-test was calculated and was above 80%, indicating that the sample is sufficiently powered to detect a difference between the groups. 80% is conventionally the acceptable level of power.
- You said that "...Each calibrated operator made three impressions with each of the OISs according to the manufacturers' recommendations." What exactly are these recommendations? Are they the same for each scanner?
Each calibrated operator took three impressions with each of the OISs in accordance with the manufacturers' instructions as described on the website of each scanner. These references were included in the manuscript.
- A Quick Start Guide for the First Scan Using the I500 Available online: https://support.medit.com/hc/en-us/articles/360042009112-A-quick-start-guide-for-the-first-scan-using-the-i500
- Carestream DENTAL CS 3600 User And Installation Manual (Page 8 of 22) | ManualsLib Available online: https://www.manualslib.com/manual/2313009/Carestream-Dental-Cs-3600.html?page=8
- Caméra Primescan | Dentsply Sirona France Available online: https://www.dentsplysirona.com/content/dentsply-sirona/fr-fr/decouvrez-nos-produits/impression-numerique/primescan.html
- I am referring here more to the scanning sequence, in terms of directions and not the steps described below.
The scanning sequences in term of directions was added as recommended by reviewer.
The scanning started from the occlusal surface from the right molar sector to the left molar sector (in red). moving to buccal surfaces of the edentulous ridge with a reverse path (orange)
and finally, the palatal surfaces of the edentulous ridge from the right to the left molar sector (yellow)
Each of the three calibrated operators took three impressions with each of the OISs according to the manufacturers' instructions [12–14]. A total of thirty-six digital impressions (nine per camera) were made. These impressions provided the digital models needed for the evaluation.
Two different scanning methods were applied as suggested by the manufacturers [12–14]. The first method is common to the Trios 4, Primescan and i500 scanners. It includes a first scan of the reference digital model without scan bodies. The scan started with the occlusal surface from the right molar sector to the left molar sector, then proceeded to the buccal surfaces of the edentulous ridge in a reverse path and finally to the palatal surfaces of the edentulous ridge from the right molar sector to the left molar sector (Figure 8). The second scan was performed with the scan bodies in place, after the circular cut and implant areas removal.
The second method was performed with the CS 3600 scanner. The first step of the scan was performed without the scan bodies and was identical to the previous method (Figure 7). Indeed, after the circular cutting of the implant areas, one scan body was placed and selected for scanning. Then, the scan body was removed and the next one was placed for scanning until all scan bodies were completely scanned on the 3D model.
Figure 8: Directions of the scanning sequence
- What are implants 1, 2, 3, 4 and 5? Not identified!
Implant 1 corresponds to the implant located in the position of the right maxillary first molar
Implant 2 corresponds to the implant located in the position of the maxillary right canine
Implant 3 corresponds to the implant located opposite the maxillary right central incisor
Implant 4 corresponds to the one positioned opposite the maxillary left canine
Implant 5 corresponds to the one positioned opposite the maxillary left first molar
Scan bodies 1, 2, 3, 4 and 5 correspond to implants 1, 2, 3, 4 and 5 respectively.
Results
- According to what is illustrated in Figure 1, you superimposed 3 digital models, right?
So, it is necessary to explain better what they obtained as a result.
The results obtained are explained
- What, exactly, do you call a "reference"?
The reference corresponds to the digital reference model
Discussion
- The Discussion should be rewritten contemplating some of the suggestions made above and making considerations about important points that generate doubts.
The discussion section was rewritten.
For example: what is the impact of model scans for the analyzes being done? Can't benchtop (scanner de table) scanning also have its own problems? In addition, it deepens the considerations about extraoral scanning (scanner de table) with intraoral scanners, which are quite different things.
Benchtop scanners have an accuracy that ranges from 5-30 µm, using as many cameras to capture the topography of the model or impression. When provided with an accurate impression, technicians can scan the impression or poured model or work directly with digital impression scan data to efficiently produce highly accurate margins, natural tooth morphology and anatomy, as well as anatomical substructures and abutments. These technological developments paralleled advancements in increased accuracy of implant restorations and abutments.
The benchtop scanner offers the ability to fabricate accurate and complex cases that still exhibit the highly esthetic and delicate dental art of traditionally produced restorations in a more efficient manner and one that also serves to digitally document each case and the materials used. Just as intraoral scanners in a clinician’s office will only be as good as the operator, benchtop scanners will only be as good as the technicians who use them.
For intraoral scans, there is a variation in the focal plane. Indeed, the distance between the object (teeth, implants, mucosa) and the tip of the scanner varies according to the practitioner. It depends on the operator when taking the impression.
On the other hand, extra-oral scanners allow a better precision because the object (cast) is placed in the scanner. A software will allow a precise and reproducible scanning. It does not depend on the operator and the scanner camera does not move. The result is therefore more reliable.
The passivity of the bridge framework is required for the long-term survival of the implants.
Reference: Jansen Curtis E. Understanding the Potential of Digital Intraoral and Benchtop Scanning Workflows. Updates in Clinical Dentistry. August 2017
If research hypotheses are formulated, each one should be accepted or rejected and discussed.
The formulated research hypotheses were rejected.
We added the results of the formulated hypotheses in the Discussion section as follow.
Our study aims to compare the accuracy of four intraoral scanners and one conventional impression for full-arch implant-supported prostheses. The first null hypothesis that conventional and digital impressions will produce casts of similar trueness was rejected. As a result, the mean absolute distance deviation and the direction of the distance deviation from the reference digital model were reduced for the conventional impression. Similarly, the I-500 had the lowest angular deviation, followed by the Trios 4 and the CS3600. The second null hypothesis, that the conventional and digital impressions will produce casts of similar precision, was also rejected. Thus, the conventional and I-500 impressions showed the lowest dispersion of values around their means. The calculated power of the post-tests was greater than 80%, indicating that the sample was sufficiently powerful to detect a difference between the groups.
We have also discussed in the Discussion section the null hypotheses rejected.
Regarding the limitations, I also understand that they can be better evaluated.
The limitations of our study were evaluated in the Discussion section.
It is worth noting that the results of this in vitro study do not presume the clinical validity of the impressions. Clinical studies are needed to evaluate the accuracy of these impressions in implant-supported complete prosthesis, as many important parameters (mouth opening, presence of blood or saliva, anatomical obstacles, ...) may affect their accuracy. However, in vitro experiments have the advantage of limiting confounding parameters and evaluating all IOS under the same conditions. In this context, it is important to follow the manufacturer's instructions, calibrate accurately, and change the tip regularly to take full advantage of the IOS's performance. In addition, the use of the mesh/mesh (virtual model) method to evaluate all the impressions may also be a limitation of our study. Meshes are surface reconstructions, and thus geometric approximations of the scanned model, which may induce errors in the calculation of distances between scan bodies. Though, in the case of implant prostheses, the first step in CAD is to replace the scan body meshes with the corresponding scan body library file. This is a geometrically perfect file (NURBS file, Non-uniform Rational B-Splines). The use of these NURBS files allows to obtain more reliable linear distances.
Conclusion
- Regarding the conclusion, as presented it looks more like a discussion paragraph, besides not corresponding to what appears in the Abstract.
The conclusion section was adjusted as recommended by reviewers.
Within the limits of the in vitro study, our results revealed that the conventional impression was more accurate than the digital impression, but further clinical studies are needed to confirm these results. Nevertheless, the continuous progress of intraoral scanning technologies and the development of new acquisition processes may allow optical impressions to extend its indications in implantology and match or even surpass conventional impressions for implant-supported complete prosthesis.
References
- Reference #15 is too old (1991)!
We added the following new reference.
Abduo, J. Accuracy of Casts Produced from Conventional and Digital Workflows: A Qualitative and Quantitative Analyses. J Adv Prosthodont 2019, 11, 138–146, doi:10.4047/jap.2019.11.2.138.

Round 2
Reviewer 2 Report
Thanks for extensive review in the manuscript.
Author Response
Thank you for your feedback.
